# Natural Drugs: A New Direction for the Prevention and Treatment of Diabetes

**DOI:** 10.3390/molecules28145525

**Published:** 2023-07-20

**Authors:** Peishan Wu, Xiaolei Wang

**Affiliations:** Endocrine and Metabolic Diseases Hospital of Shandong First Medical University, Shandong First Medical University & Shandong Academy of Medical Sciences, Jinan 250001, China; 15564891550@163.com

**Keywords:** mitochondrial stress, insulin resistance, natural drugs

## Abstract

Insulin resistance, as a common pathological process of many metabolic diseases, including diabetes and obesity, has attracted much attention due to its relevant influencing factors. To date, studies have mainly focused on the shared mechanisms between mitochondrial stress and insulin resistance, and they are now being pursued as a very attractive therapeutic target due to their extensive involvement in many human clinical settings. In view of the complex pathogenesis of diabetes, natural drugs have become new players in diabetes prevention and treatment because of their wide targets and few side effects. In particular, plant phenolics have received attention because of their close relationship with oxidative stress. In this review, we briefly review the mechanisms by which mitochondrial stress leads to insulin resistance. Moreover, we list some cytokines and genes that have recently been found to play roles in mitochondrial stress and insulin resistance. Furthermore, we describe several natural drugs that are currently widely used and give a brief overview of their therapeutic mechanisms. Finally, we suggest possible ideas for future research related to the unique role that natural drugs play in the treatment of insulin resistance through the above targets.

## 1. Introduction

Diabetes mellitus is associated with metabolic disorders, with 25% of patients progressing to several microvascular and macrovascular complications, followed by blindness, renal failure, myocardial infarction, or stroke within 20–40 years [1]. Diabetes mellitus can also progress to diabetic ketoacidosis in 10% of cases, which is the leading cause of diabetes mellitus-related death worldwide. As type 2 diabetes (T2DM) is a major burden on healthcare systems, its mechanism has been widely studied for many years, and the most important mechanism leading to T2DM has been shown to be insulin resistance [2]. The term insulin resistance refers to the decreased efficiency of insulin in promoting glucose uptake and utilization in tissues involved in glucose homeostasis for various reasons, for which the body compensates by secreting too much insulin to produce hyperinsulinemia to maintain the stability of blood glucose. Patients with insulin resistance often have other abnormal states, such as impaired glucose tolerance (IGT) and impaired fasting glucose. Glucose tolerance depends on the insulin secretion of the pancreas on the one hand and on the target organ’s sensitivity to insulin, both of which determine whether insulin can work at the right time. When the body develops impaired glucose tolerance, the risk of developing diabetes is greatly increased [3]. Some clinical studies have shown that muscle insulin resistance is more obvious than liver insulin resistance in IGT subjects, and insulin secretion is significantly impaired during the whole period of diabetes [4]. Decades of extensive research results and clinical trials have provided a detailed protocol for the treatment of these diseases, but the specific mechanism remains unclear, greatly limiting early intervention and the prevention and treatment of complications.

A growing recognition of the integrality of metabolic physiology has led to research into the mechanisms and related factors influencing insulin resistance. As early as 1936, Himsworth clarified the concept of insulin resistance. Initially, it was assumed that reduced insulin receptor binding was responsible for the typical obesity-related insulin resistance, but this hypothesis was soon replaced by a model centered on insulin signal transduction defects. The mechanism of insulin resistance is complex, and its pathological changes can also lead to a series of diseases. Most of these diseases are caused by a combination of insulin resistance in skeletal muscle, liver, and fat. Insulin resistance [5] significantly impairs insulin-induced glycogen synthesis in the liver and glucose uptake in muscle and fat. In addition, adipose insulin resistance can lead to excessive triglyceride levels in the liver of patients with non-alcoholic fatty liver disease by promoting the reesterification of circulating fatty acids.

Various research groups have developed new drugs based on mechanisms that have been discovered in recent years. For instance, in 2005, Burkey et al. [6] realized that peptidyl peptidase-IV can improve insulin resistance and help in the treatment of diabetes. Regarding the crosstalk between insulin-responsive tissues, which is one of the complex mechanisms associated with insulin resistance in vivo, Klymenko et al. [7] have carried out relevant expositions and experiments over the years. Traditional thiazolidinediones (TZDs) or metformin, sodium-dependent glucose transporter 2 (SGLT-2) inhibitors, and other medications can have undesirable side effects [8,9,10], while natural drugs have shown unique advantages due to their wide range of targets, few side effects and mild efficacy. Many studies have shown that natural drugs such as cassia semen, rhubarb, aloe vera, and senna can improve insulin resistance [11,12,13].

As early as 2016, Jeong, E. M.’s group [14] demonstrated the beneficial effects of ameliorating mitochondrial oxidative stress on insulin resistance and diastolic dysfunction by treatment with the mitochondria-targeting antioxidant MitoTEMPO. Liver, muscle, and fat are the three major organs associated with peripheral insulin resistance, among which muscle and fat insulin resistance have been pointed out to be closely related to mitochondrial dysfunction [15,16]. The liver is the starting point of insulin resistance and plays a central regulatory role in glucose and lipid metabolism [17]. Moreover, the liver is rich in mitochondria, which, as the “energy factory” of cells, are the main site where reactive oxygen species (ROS) are produced. When ROS are produced faster than they can be removed by the body [18], mitochondrial stress and energy metabolism disorders follow, and they play an important role in the early onset of a variety of metabolic diseases [19,20]. This makes it possible to further explore the specific mechanisms of hepatic insulin resistance.

Overall, in our review, we will introduce the newly discovered cytokines and genes identified by various research groups that are closely related to mitochondrial stress and insulin resistance. This review will also provide a unique summary of some natural drugs for the treatment of insulin resistance-related targets to pave the way for future research directions (Figure 1).

Natural drugs are an important means of prevention and treatment of insulin resistance-related diseases such as diabetes. Insulin resistance is closely related to many pathological processes, such as abnormal insulin signaling pathway conduction, energy metabolism regulation, and inflammation, and emodin, aloe-emodin, chrysol, and beranin, as the main active components of many natural drugs, play a role by targeting one or more targets in the above pathological processes.

## 2. Mitochondrial Stress, a Key Inducer of Insulin Resistance

Mitochondria have attracted increasing attention due to their complex functions, unique genome, and important role in energy metabolism [21]. In the physiological state, mitochondria regulate cell apoptosis and maintain body homeostasis through their own replication, translation, fusion and fission, mitophagy, and other important factors [22,23]. Under pathological conditions, mitochondrial dysfunction damages cellular homeostasis and creates conditions for the occurrence of a variety of diseases [24,25]. The prevailing view is that mitochondrial defects can be caused by mutations in the mitochondrial genome or by chronic exposure to proinflammatory cytokines, including type I interferon [26]. Mutations in the mitochondrial genome often lead to the appearance of defective proteins and mis-targeted proteins in the mitochondrial proteome, thereby mediating the inactivation or overactivation of the proteasome. In an abnormal state, the damaged homeostasis of the body will first seek to correct itself. For example, mitophagy, which develops from mitochondrial fission, is a way for the body to remove abnormal mitochondria [27]. When mitochondrial dysfunction exceeds the compensatory limit of the body, mitochondrial stress follows, which is manifested as a decrease in the mitochondrial inner membrane electrochemical (IM) potential mainly through the activity of the respiratory chain and the accumulation of ROS caused by dysfunction of the respiratory chain [28]. Mitochondrial self-repair and programmed death are not two independent processes. In the study by Wu, B. et al. [29], it was observed that energy deficiency caused by decreased mitochondrial ATP synthesis stimulates *AMP-activated protein kinase* (*AMPK*), which partially restores mitochondrial energy metabolism and biogenesis but also induces mitophagy. Overall, the origin of mitochondrial defects is complex, and if uncorrected, these defects can progress to the next stage of mitochondrial stress.

As mentioned earlier, the pathological process of insulin resistance in the liver is closely related to mitochondrial stress. Mitochondrial stress and insulin resistance form a vicious cycle that jointly promotes the development of the disease. In recent years, people have gradually turned their attention toward the mechanism of mitochondrial stress, hoping to treat metabolic diseases with this as a target. In Figure 2, we visualize some of the factors that play an important role in mitochondrial stress and insulin resistance and point out some possible bridging factors between the two.

### 2.1. Major Factors Leading to Mitochondrial Stress

Many research groups worldwide have discussed the mechanisms involved in mitochondrial stress. ROS, the mediators of mitochondrial stress, have long been thought to play an important role in neurodegenerative diseases and cardiovascular diseases. As early as 1994, Paolisso et al. [30] uncovered a relationship between mitochondrial stress and metabolic diseases by measuring O_2_ levels in the bodies of non-insulin-dependent diabetic patients. Over the next two decades, various research teams have proposed several related factors that affect mitochondrial stress. Among them, the most representative and most likely therapeutic targets include PPARs, CypD, and AGEs, which we will discuss in the following sections.

PPARs, activated receptor ligands of the nuclear hormone receptor family that are classified into three subtypes and have been found to control many intracellular metabolic processes, are ligand-inducible nuclear receptors. PPAR-*β/δ* inhibits mitochondrial stress and reduces ROS production on the one hand and reduces inflammation and insulin resistance induced by interleukin-6 on the other hand, thereby delaying the disease progression of obesity, T2DM, NASH, and other diseases [31,32]. In 2015, Lee et al. [33] noted that PPAR-β/δ of the PPAR family has remarkable effects on improving hepatic insulin sensitivity and reducing insulin resistance and blood sugar levels. Knockout or low expression of PPAR-γ significantly reduced macrophage activation, putting mice at increased risk of obesity and insulin resistance. In summary, PPARs play a role in fatty acid decomposition, cholesterol transport, and energy metabolism, suggesting that PPARs can be an important treatment option for some diseases.

Some studies have mentioned that the key molecule CypD plays an important role in regulating mitochondrial stress. The mitochondrial permeability transition pore (MPTP) is a nonselective channel in the mitochondrial intima. Under physiological conditions, periodic opening and closing are essential for maintaining cell homeostasis [34]. However, an abnormal increase in or modification (acetylation) of the protein expression of the key mitochondrial stress protein CypD leads to excessive irreversible opening of the MPTP [35], causing mitochondrial swelling and dysfunction of the mitochondrial respiratory chain (impaired activity of mitochondrial complex enzymes I–V) and leading to mitochondrial stress and energy metabolism disorders. In 2019, Castillo et al. [36] showed that after CypD caused MPTP opening, the presence of nonesterified fatty acids led to proton leakage from the islets. Changes in CypD expression or acetylation are considered key factors in the regulation of mitochondrial stress; protein acetylation modification, as one of the important aspects of epigenetics, has been a hot research topic in recent years. Recently, we found that in an insulin resistance mouse model induced by a high-fat diet, the luciferase activity and protein expression of CypD in the liver were increased compared with those in the control group [18]. Moreover, CypD gene knockout reduced liver mitochondrial stress, thereby alleviating insulin resistance.

Two molecules that affect mitochondrial stress have thus far been described. Therefore, let us look at the compounds involved. AGEs are stable covalent compounds generated by the spontaneous reaction of macromolecular substances such as proteins, lipids, or nucleic acids with glucose or other reduced monosaccharides without the involvement of enzymes. Numerous research teams have found that AGEs accelerate aging in the human body; that is, they are closely linked to the development of numerous chronic degenerative diseases [37]. The AGE-RAGE axis activates NADPH oxidases, which are the major endogenous sources of ROS, thus aggravating the occurrence of mitochondrial stress [37]. The above process mainly occurs through the mitochondrial respiratory chain and occasionally occurs through stress-related signaling pathways, such as Jun N-terminal kinase and p38. NF-κB activated by AGEs also increases the expression of certain cytokines, such as those that promote inflammation, and the increase in inflammation in the body leads to the exacerbation of disease [17]. AGEs have long been used as biomarkers or predictors of diabetes complications; however, in 2020, Shen et al. [38] showed that the promotion of ROS by AGEs implies that AGEs have a role in the treatment of diabetes, NASH, and other diseases.

In addition to these three factors, other molecules and compounds that cause mitochondria to produce too many ROS are being discovered. *Cytochrome P450 family 2 subfamily E member 1* (*CYP2E1*), located on chromosome 10, is mainly involved in the metabolism of low-molecular-weight substances. High insulin levels in the body and other pathological changes, such as steatosis, can promote the body’s ability to increase the production of ROS [39]. The oxidizing capacity of iron also contributes to the production of ROS [40]. In addition, some recent studies have shown that the activation of NADPH oxidase 4 (NOX4) promotes ROS production while prolonging the endoplasmic reticulum stress mediated by OS, which leads to liver cell apoptosis and accelerates the progression of NASH and other diseases [41]. In addition, many experiments have shown that liver-specific *PGC-1* deficiency expands the influence of mitochondrial stress and accelerates the development of NASH in mice, suggesting that PGC-1 inhibits oxidative stress and inflammation, leading to improvement in disease [42]. In terms of compounds, in 2016, Chakraborti et al. [40] showed that a diet rich in fructose greatly speeds up the production of ROS.

These factors usually regulate insulin resistance by affecting mitochondrial stress, but some factors not only mediate or inhibit mitochondrial stress but also aggravate or improve insulin resistance themselves. For example, many scholars have demonstrated that microRNAs can reduce insulin resistance either directly [43] or indirectly (by inhibiting mitochondrial stress and so on) [44]. MicroRNAs, or miRNAs for short, are a class of endogenous small noncoding RNAs with regulatory functions found in eukaryotes. Several articles have described the mechanisms of these small RNAs in various types of diabetes and other metabolic diseases [45,46]. *miR-155-5p* protects beta cells from insulin resistance by promoting insulin secretion by the body’s beta cells in the presence of glucose [47]. Moreover, *miR-126* and *miR-30c* are two important benign factors in patients with diabetic cardiomyopathy. The former indirectly activates sirtuin-1 (SIRT1) and superoxide dismutase (SOD) to induce resistance to oxidative stress, while the latter targets PGC-1, not only inducing PPAR-α but also playing a positive role in key mitochondrial regulation. Together, the two can reduce excessive ROS and myocardial lipid accumulation in patients with diabetic cardiomyopathy, thereby preventing pathological processes such as cardiac insufficiency [48]. miRNAs have been applied in various fields because of their newly discovered wide use. At present, most in vitro experiments on reducing the effects of insulin resistance and treating diabetes with miRNAs have been successful, but the therapeutic effects of miRNAs in vivo remain to be further studied.

### 2.2. How Does Mitochondrial Stress Affect Insulin Resistance?

Based on some clinical studies, a variety of drugs have also been observed to act by targeting mitochondrial dysfunction. Empagliflozin (EMPA) [49] and pyridostigmine (PYR) [50] have good preventive effects on diabetic heart damage because they can regulate the shape and function of mitochondrial cristae, thereby improving mitochondrial function in heart tissue, and can regulate mitochondrial fusion and fission. Wang, X. et al. [51] observed that zinc finger, BED-type containing 6 (ZBED6) may target the abnormal accumulation of mitochondrial ROS, thereby improving insulin resistance in the body. The therapeutic effects of some natural medicines on diabetes are also dependent on the regulation of mitochondrial function, including astaxanthin (AX) treatment, which stimulates mitochondrial biogenesis by activating the AMPK pathway in skeletal muscle and significantly improves insulin resistance [52]. Many studies have confirmed that the relationship between glucose tolerance, insulin resistance, and mitochondrial dysfunction may not be direct. Factors that play a role in this process would be effective targets for drugs to prevent and treat diseases related to insulin resistance, which is why investigating how mitochondrial stress leads to insulin resistance has been the focus of many research groups.

#### 2.2.1. HSP60

Heat shock protein 60 (HSP60) is a chaperone protein that is overexpressed when mitochondrial stress occurs. This state not only impairs insulin signaling and insulin sensitivity in mouse liver cells but also promotes endoplasmic reticulum stress, which induces liver adipogenesis and insulin resistance. In 2020, Xiao et al. [53] presented experimental evidence for this claim. In their experiment, HSP60 knockout inhibited the mammalian target of rapamycin complex 1—sterol-regulatory-element-binding protein 1 (mTORC1-SREBP1) signaling and hepatic adipogenesis, suggesting that HSP60 may be involved in endoplasmic reticulum stress-induced mTORC1-SREBP1 signal transduction related to adipogenesis and steatosis. Therefore, the researchers concluded that mitochondrial stress and endoplasmic reticulum stress coregulate the insulin resistance process in the body.

However, HSP60 acts differently in different parts of the body. On the one hand, as described previously, *miRNAs* play an important role in influencing insulin resistance. On the other hand, a high serum level of HSP60 is often accompanied by the downregulation of HSP60 in the myocardium, which means that the inhibition of HSP60 in the myocardium will lead to the intensification of insulin resistance [54]. In 2020, Wen et al. [38] found that miR-802-5p causes cardiac insulin resistance by reducing HSP60 expression. HSP60 is also an important therapeutic target for insulin resistance in the heart, as researchers have recently discovered.

#### 2.2.2. FGF21

Does mitochondrial stress under pathological conditions necessarily result in insulin resistance? In 2013, Kim argued against this. *Autophagy-related 7* (*ATG7*) is a gene that is critical for autophagosome activity; in the *ATG7* knockout mouse, Kim et al. [55] were surprised to observe that the mice were not fat but lean and were unaffected by obesity and insulin resistance induced by a high-fat diet. Inhibition of autophagy leads to mitochondrial stress; however, this mitochondrial stress improves insulin resistance. They further explored the bridging factor between the two. In a series of experiments, they identified a factor called fibroblast growth factor 21 (FGF21), which once served as a marker for a variety of mitochondrial diseases caused by mutations in mitochondrial DNA (mtDNA) [56,57]. Researchers have found that inhibiting the expression of mitochondrial oxidative phosphorylation genes results in damage to the mitochondrial respiratory chain and a reduction in ATP, thereby activating the integrated stress response (ISR) and mediating the production of FGF21 [58,59]. Similar to the HSP60 knockout results, FGF21 also improves hepatic insulin sensitivity by inhibiting mTORC1. Moreover, FGF21 activation increases fatty acid oxidation, lipolysis, and the browning of white adipose tissue in mice, which protects mice from obesity and insulin resistance. The researchers also observed that metformin administration increases serum levels of FGF21, suggesting that FGF21 is induced by metformin, providing indirect evidence of FGF21’s therapeutic value [60].

Currently, FGF21 and its analogs have been used in laboratory animals and human subjects to significantly improve lipid distribution in the body and to reduce the body weight of subjects, which is why it has become a promising new drug for the treatment of obesity, diabetes, and metabolic syndrome. Unfortunately, in addition to the mitochondrial stress mentioned above, FGF21 can also be induced by a variety of stresses, such as obesity, exercise or cold exposure, and the PPAR family. The connection between these obvious nonmitochondrial inducers of FGF21 and mitochondrial stress is still unclear and needs to be uncovered.

#### 2.2.3. GDF-15

Stress response protein growth differentiation factor-15 (GDF-15) is secreted during mitochondrial stress or functional damage [61,62,63], and high levels of GDF-15 are closely associated with a variety of pathological diseases, including inflammation, cancer, and various mitochondrial diseases [64], which is why GDF-15 has been extensively explored as a biomarker of mitochondrial disease and is thought to be closely related to the mitochondrial function of patients [65,66,67]. The increase in GDF-15 in the body has been proven to be induced by dysfunction of the mitochondrial respiratory chain rather than other metabolic dysfunctions [68]. Furthermore, Straub [69] suggested a potential connection between mitochondrial respiratory dysfunction, the NADH/NAD ratio, impaired ATP levels, and GDF-15. Steffen, J. et al. [70] showed that knockout of drp leading to ER stress and mitochondrial abnormalities can activate the AFT4-controlled integrated stress response (ISR) and increase the hepatic expression of GDF-15, which subsequently aggravates inflammation, fibrosis, and necrosis. This result implies that GDF-15 regulates its own expression in response to impaired mitochondrial function through the ISR pathway, just as the ISR pathway also induces GDF-15 hyperexpression during metformin treatment. Treatment with GDF-15 and metformin induces a process that is also related to multiple pathways. Aguilar-Recarte et al. [71] showed that metformin can maintain full AMPK activation by upregulating GDF-15 to participate in the therapeutic process of this drug. The study by Zhang, S. Y. [72] also observed that metformin regulates energy homeostasis through the renal GDF-15-dependent AP axis. In conclusion, the above results indicate that mitochondrial stress caused by mitochondrial dysfunction induces an increase in GDF-15. GDF-15 can not only be used as a marker of some metabolic diseases but can also play an important role in the diagnosis and treatment of diseases such as diabetes mellitus.

The relationship between GDF-15 and insulin resistance has also been explored by some research groups. Several studies in recent years have found that GDF-15 can be used as a therapeutic target for obesity and prediabetic glucose tolerance disorders. Studies have shown that GDF-15 is associated with obesity and insulin resistance, abnormal blood glucose levels, and impaired blood glucose homeostasis in the body. In 2017, Chung et al. [73] used recombinant GDF-15 in ob/ob mice and found that the mice lost weight and exhibited improved insulin sensitivity. In 2020, Choi [62] found that GDF-15 and FGF21 secretion was significantly upregulated in adipocyte-specific Crif1 (also known as Gadd45gip1) knockout mice on a high-fat diet, which was able to inhibit weight gain and improve glucose tolerance. Recently, the relationship and the specific mechanism between GDF-15 and insulin resistance have received increasing attention.

GDF-15 shows complex effects in different stages of disease; however, its specific mechanism in the early onset of metabolic diseases, especially hepatic insulin resistance, is still unclear. As we mentioned earlier, CypD plays an important role in the regulation of mitochondrial stress, and there is a large body of evidence to suggest this; at the same time, our group’s previous study suggested that GDF-15 is downstream of CypD. Therefore, in this review, we propose a potential mechanism that describes the possible relationship between CypD and GDF-15. Increased expression or activity of CypD causes mitochondrial stress and energy metabolism disorders, which may affect insulin resistance by regulating the stress response protein GDF-15. The specific mechanism remains to be further studied (Figure 3).

## 3. Complex Mechanisms of Natural Drugs in the Treatment of Diseases Related to Insulin Resistance

Plant-derived secondary metabolites [74] are organic compounds produced by plants and are commonly known as a class of natural drugs. Plant-derived secondary metabolites are more readily available and safer than conventional chemical drugs [74] and have been shown to have clinically meaningful and mild efficacy in cancer treatment, inflammation reduction, and diabetes treatment. To date, researchers have identified more than 400,000 secondary metabolites in nature [75], including allicin, quercetin, eugenol, lycorine, tea polyphenols, and berberine, and more new plant-derived active compounds are being discovered. Among them, these active compounds can be widely classified into terpenoids and their derivatives, alkaloids, steroids, amino acids, polysaccharide antimicrobial peptides, lignans, saponins, and new structures [75]. Recently, an increasing number of research teams have begun to pay attention to the effect of secondary metabolites on insulin resistance and related mechanisms in the pathogenesis of diabetes.

Many studies have confirmed the good hypoglycemic ability of a variety of plant derivatives. Moreover, numerous clinical cases have verified the therapeutic effect of resveratrol on diabetic patients [76]. Fasting blood glucose and serum insulin levels were significantly reduced during the treatment, and insulin sensitivity and glucose and lipid metabolism of the body were also restored [77]. Similar to the way that resveratrol exerts its therapeutic effect, curcumin, as a new drug for obesity and diabetes, can also alleviate the progression of insulin resistance through AMPK, nuclear factor erythroid 2-related factor 2 (Nrf2) and PPAR-related molecular pathways [78,79], which may involve endoplasmic reticulum stress and other oxidative stress abnormalities [80]. This process is closely related to the regulation of the intestinal flora [81]. However, anthocyanins (anthocyanin 3-glucoside) and proanthocyanidins (PCs), which are widely present in black bean seeds, have been shown in animal experiments to activate AMPK- and glucose transporter type 4 (GLUT4)-related pathways, alleviate hyperglycemia, and restore insulin sensitivity in mice [82]. In addition to the AMPK-related pathway, targeting other important factors in the regulation of insulin resistance can achieve similar therapeutic effects. The abnormal activation of IRS often leads to impairment in the insulin signaling pathway and then mediates the occurrence of insulin resistance through inflammatory factors such as tumor necrosis factor (TNF)-α [83]. Carnosic acid and rosmarinic acid can reduce the occurrence of insulin resistance by regulating the AMPK pathway and IRS activation [84,85,86]. In addition, Carica papaya extracts are equally rich in plant polyphenols. It has been confirmed that they reduce blood glucose and lipid levels, and the mechanism may be related to the improvement in endothelial NO synthase (eNOS), which is related to the accumulation of ROS and energy imbalance [87]. In summary, plant polyphenols are beneficial in the treatment of insulin resistance, and the mechanism is often closely related to abnormal insulin signaling mediated by oxidative stress.

Anthraquinone natural products, including emodin, aloe-emodin, rhein, chrysophanol, aurantio-obtusin, and alaternin, are the main active ingredients of many natural drugs. Existing studies have found that they can alleviate insulin resistance from many aspects, such as regulating the insulin signaling pathway, regulating energy homeostasis, and improving inflammation. Emodin has been shown to increase the phosphorylation of AMPK protein and acetyl-CoA carboxylase (ACC) protein in the liver, upregulate the expression of carnitine palmitoyl transferase 1 (CPT1), and downregulate the expression of SREBP-1c and fatty acid synthase (FAS), thereby improving energy metabolism disorders, reducing fat accumulation, and promoting cellular glucose absorption. Emodin can reduce the levels of fasting blood glucose and fasting insulin in mice and can improve insulin resistance [88,89]. Moreover, emodin can promote M2 polarization of macrophages by increasing triggering receptor expressed on myeloid cells 2 (TREM2) expression, significantly reduce local and systemic inflammatory responses in obese mice, inhibit weight gain and lipid accumulation, and reduce fasting blood glucose and fasting insulin levels [90]. Aloe-emodin has also been found to be effective in reducing the production of the inflammatory factors TNF-α and interleukin (IL)-6 and in suppressing the NF-κB signaling pathway, thereby restoring insulin signaling, lowering fasting blood glucose, regulating islet β-cell function, and inhibiting fat accumulation to reduce obesity and insulin resistance. Aloe-emodin has been shown to alleviate inflammation [13,91,92,93]. Rhein can also improve insulin resistance by reducing adipose tissue inflammation and liver triglyceride accumulation in mice [94]. In addition, rhein promotes macrophage polarization toward the M2 phenotype [95], which can reduce inflammation and thus alleviate insulin resistance [96]. Cassia seed, a natural drug rich in anthraquinone products such as aurantio-obtusin and alaternin, can reduce the fasting blood glucose and fasting insulin contents in obese mice and restore insulin sensitivity by increasing skeletal muscle glucose uptake in obese mice [97]. The targets of aurantio-obtusin are more diverse. On the one hand, it phosphorylates AMPK, an important pathway for energy homeostasis. On the other hand, the expression of FAS is inhibited to reduce fat synthesis and thus improve insulin resistance [98]. In addition, aurantio-obtusin can activate the phosphoinositide 3-kinase (PI3K)–protein kinase B (AKT) signaling pathway in liver and adipose tissue, reduce fasting blood glucose, and improve glucose tolerance [99]. Alaternin also plays a role in regulating insulin signaling. Alaternin regulates downstream signals of insulin and restores insulin signaling by competitive inhibition of protein tyrosine phosphatase 1B (PTP1B) [100].

Among natural drugs, other natural products besides plant polyphenols and anthraquinones have shown promising results in the treatment of insulin resistance. Allicin, an organic sulfur compound extracted from the bulb of Allium garlic, has been shown to be effective in the treatment of various diseases due to its specific actions, including antibacterial, antioxidative stress, regulation of cardiovascular and cerebrovascular diseases, reduction in blood lipids, and regulation of diabetes. One study showed that allicin had a hypoglycemic effect in type 2 diabetes patients and animal models in which it could activate pancreatic insulin secretion and reduce total cholesterol (TC), triglyceride (TG), and fasting blood glucose (FBG) levels [101]. Zhai et al. [102] showed that the hypoglycemic activity of allicin in diabetic rats was similar to that of glibenclamide and insulin. In addition, similar to the way in which allicin acts, allyl propyl disulfide, cysteine sulfoxide, and S-allyl cysteine sulfoxide prevent liver-induced insulin activation, increase the insulin production capacity of islet β-cells, separate insulin from its bound form, and increase the insulin sensitivity of cells. The subsequent effect is to reduce blood glucose levels [103,104]. Therefore, the use of bioactive compounds derived from natural sources has become a new direction for the treatment of diabetes due to the moderate and obvious therapeutic effects of these metabolites, which is very consistent with the idea of treating chronic diseases.

The side effects caused by drugs in the process of disease prevention have always been the focus of academic attention, which is why early intervention for diabetes is basically based on exercise for prevention and dietary intervention. Based on the low side effects and high safety characteristics of natural medicines, we speculate that natural medicines play a key role in disease prevention and are expected to be used as an adjunct to prevention methods such as exercise [105,106]. In addition, the treatment of patients with gestational diabetes mellitus requires that close attention be given to the effects of drugs on the mother and fetus, as natural drugs may also have unexpected effects. We further hypothesized that oral natural medicines might improve glucose metabolism through the regulation of the gut microbiota, and a recent study preliminarily confirmed our hypothesis [107]. We also found that Leyva-Gomez, G. et al. [108] proposed the use of natural medicine-rich films to promote wound healing, which may be a new application for the prevention and treatment of diabetic foot. Although the extraction of some natural products is limited by technology and cost [109], we believe that natural products will become new stars in various fields.

## 4. Conclusions and Future Perspectives

In this article, we introduced the mechanisms of mitochondrial stress and insulin resistance and the high-profile inducing or inhibiting factors, and importantly, we proposed that CypD may lead to abnormal mitochondrial stress and affect insulin resistance through GDF-15, which represents a direction for future research (Table 1).

Currently, an increasing number of drugs have been applied to the treatment of diseases related to glucose and lipid metabolism disorders. Throughout years of exploring the efficacy and safety of various diabetes drugs, natural drugs have shown unique advantages. We thus put forward a good prospect: In the future, natural drugs will become the new favored approach in the process of diabetes prevention and treatment due to their high safety profile and broad action targets. Plant polyphenols, in particular, are closely related to oxidative stress and alleviate the progression of diabetes caused by insulin resistance through the regulation of the insulin signaling pathway. We look forward to more research teams discovering additional natural drugs, and we hope to further explore the specific mechanisms by which natural drugs work, which represents a research direction with great potential.

## Figures and Tables

**Figure 1 molecules-28-05525-f001:**
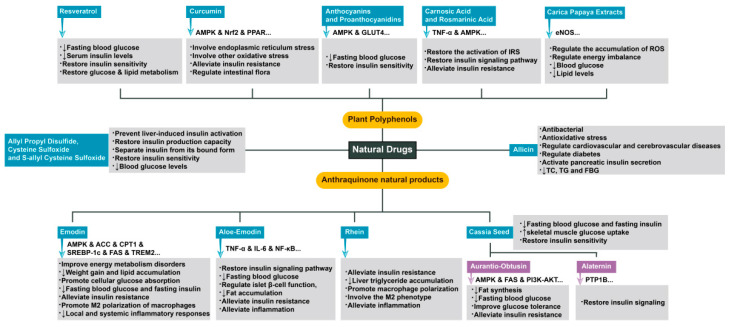
Extensive use and specific mechanisms of natural drugs in insulin resistance-related diseases. The arrows between the sections represent the preventive and therapeutic effects of the natural drug on the disease through the factors shown next to the arrows.

**Figure 2 molecules-28-05525-f002:**
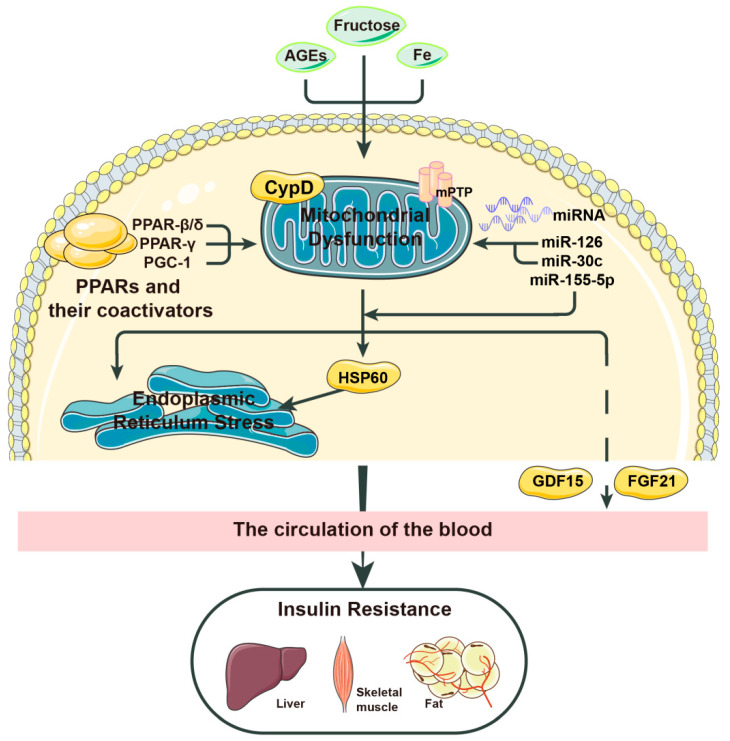
The relevant factors mentioned in this review that affect mitochondrial stress and insulin resistance or connect the two. Some pathological factors lead to increased mitochondrial stress in the body. This process can be induced or aggravated by factors such as cyclophilin D (CypD) and advanced glycation end products (AGEs), but newly discovered factors such as peroxisome proliferator-activated receptors (PPARs) and proliferator-activated receptor-γ coactivator-1 (PGC-1) can improve this pathological process. Among miRNAs, miR-30c can inhibit mitochondrial stress and insulin resistance by targeting PGC-1, miR-126 can protect the body from the effects of reactive oxygen species (ROS) imbalance, and miR-155-5p can improve the adaptation of cells to insulin resistance. Mitochondrial stress leads to an increase in ROS production, which aggravates insulin resistance in many ways, such as the overexpression of heat shock protein 60 (HSP60). However, mitochondrial stress does not only cause purely negative effects. Its activation of fibroblast growth factor 21 (FGF21) can inhibit the progression of insulin resistance to a certain extent, and we hypothesize that its activation of growth differentiation factor-15 (GDF-15) may do the same. Solid arrows indicate “direct action” with no other processes in between; The dashed arrow represents “indirect action”, which means that there are multiple action processes that have been omitted.

**Figure 3 molecules-28-05525-f003:**
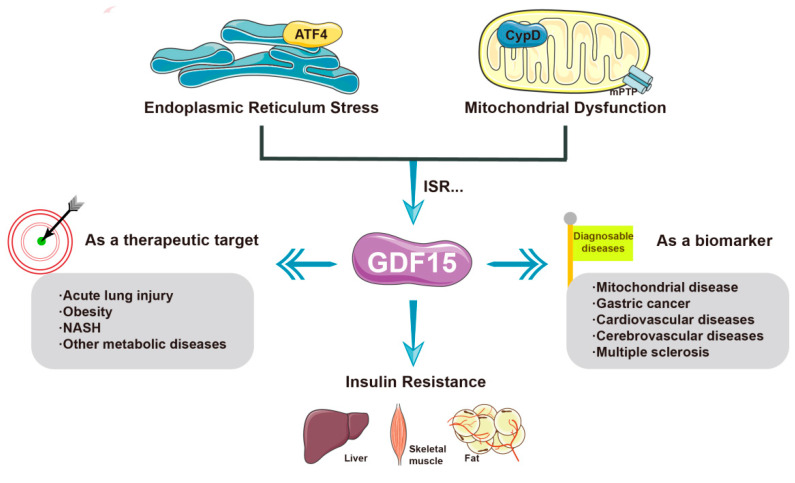
The current main applications of GDF-15 and its role in mitochondrial stress and insulin resistance. GDF-15 is an important marker in the diagnosis of many diseases. In addition, it can be used to treat the abnormal pathological state of mitochondrial stress, which is elevated to varying degrees in a variety of diseases, and its expression can effectively improve insulin resistance. For example, by activating ATF4, metformin effectively alleviates insulin resistance by inducing GDF-15. Notably, CypD may aggravate the progression of diseases related to insulin resistance by inhibiting the expression of GDF-15.

**Table 1 molecules-28-05525-t001:** A brief summary of the relevant mechanisms mentioned in the review.

Factor	Mechanism
AGEs	Promotes mitochondrial stress by activating NADPH
CypD	Promotes mitochondrial stress when the MPTP is open and NEFAs are present
PPARs	Inhibit mitochondrial stress by inhibiting IL-6
*miR-126*	Inhibits mitochondrial stress by activating SIRT1 and SOD
*miR-30c*	Inhibits mitochondrial stress by activating PGC-1 and PPAR-α
HSP60	Mitochondrial stress causes overexpression of HSP60; HSP60 promotes mTORC1-SREBP1 signal transduction and promotes insulin resistance together with endoplasmic reticulum stress
FGF21	Metformin induces mitochondrial stress through the Perk-eIF2α-ATF4 axis; mitochondrial stress promotes FGF21 expression by activating the comprehensive stress response (ISR); FGF21 improves insulin resistance
GDF-15	GDF-15 improves insulin resistance; CYPD may cause insulin resistance by inhibiting GDF-15

## Data Availability

Not applicable.

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
