# Peer review of "Natural Drugs: A New Direction for the Prevention and Treatment of Diabetes"

_molecules, 2023, doi:10.3390/molecules28145525_

Round 1

Reviewer 1 Report

Overall, the Manuscript ID molecules-2419416 appears often hard to read. The use of natural drugs for DMT2  is of  interest and has merit. I agree on the idea that natural drugs should be implemented. And suggest to focus the paper more on this topic.

The fact that mitochondrial dysfunction is involved in the development of hepatic insulin resistance is established:  dysfunctional mitochondrial respiration would impair  fatty acid β-oxidation causing hepatic fatty acid accumulation in turn establishing  the vicious circle of hepatic insulin resistance  (involving  oxidative stress). (see https://doi.org/10.3389/fphar.2019.01193).  Surely, studies addressing the role of mitochondria in insulin resistance have shown conflicting results. Mitophagy, can preserve mitochondrial function , as it can restore fatty acid oxidation improving hepatic insulin resistance. However, the idea of an involvement of the elusive MTP and CypD does not find this reviewer enthusiastic. Although CypD is a key regulator of cellular death and its inhibition may blockg the opening of mPTP its use as a therapeutic approach for mitochondria-related diseases seems too far. Instead, I find GDF-15 very interesting. 

The paper will benefit from a native speaker thorough language revision and a more direct focus on the topic of the hypothesis on the mitochondrial defect origin and the use of natural drugs , after which revision it may be worth publishing.

lines 141-143– any reference? 

add space to line 120 "pnd delays the disease progression of obesity, T2DM, 121 NASH and other conditions[18]". there are numerous errors of this type in the text.

language needs extensive editing.

see for example. line 30:

As diabetes mellitus is a major healthcare burden, the mechanism of type 2 diabetes (T2DM) has been widely discussed for many years, and the most important mechanism is insulin resistance.

 a lot of clumsy sentences such as line 152: AGEs induce the expression of their own receptors  and bind with their receptors to activate NADPH

Line 180 For example, many scholars have proven that mi- roRNAs not only inhibit mitochondrial stress to improve insulin resistance but also directly reduce the effect of insulin resistance

Hard  to read text,  line 199: Mitochondrial stress disrupts the function of pancreatic beta cells, leading to impaired glucose tolerance and insulin resistance, but the relationship between mitochondrial stress and insulin resistance is certainly not through one factor, as many factors act  as links.

Author Response

Thank you for this valuable feedback. We have studied the comments carefully and made corrections in the manuscript according to your comments.

Response to Reviewer 1 Comments

Comments and Suggestions for Authors

Overall, the Manuscript ID molecules-2419416 appears often hard to read. The use of natural drugs for DMT2  is of  interest and has merit. I agree on the idea that natural drugs should be implemented. And suggest to focus the paper more on this topic.

The fact that mitochondrial dysfunction is involved in the development of hepatic insulin resistance is established:  dysfunctional mitochondrial respiration would impair  fatty acid β-oxidation causing hepatic fatty acid accumulation in turn establishing  the vicious circle of hepatic insulin resistance  (involving  oxidative stress). (see https://doi.org/10.3389/fphar.2019.01193).  Surely, studies addressing the role of mitochondria in insulin resistance have shown conflicting results. Mitophagy, can preserve mitochondrial function , as it can restore fatty acid oxidation improving hepatic insulin resistance. However, the idea of an involvement of the elusive MTP and CypD does not find this reviewer enthusiastic. Although CypD is a key regulator of cellular death and its inhibition may blockg the opening of mPTP its use as a therapeutic approach for mitochondria-related diseases seems too far. Instead, I find GDF-15 very interesting.

Response: Thank you for your kind suggestion. Based on your interest in GDF15, we have expanded the GDF15-related content in Lines 319-322 and Lines 325-329 of the article to give it a more prominent position in the article. In addition, our research team has made advancements in CypD-related research (https://doi.org/10.1002/hep.29788), and our experimental results show that some nonspecific drugs, such as cyclosporine A, an inhibitor of CypD, can play a therapeutic role in diabetic models mediated by factors such as mitochondrial dysfunction. We also look forward to the development and use of some specific drugs targeting CypD.

The paper will benefit from a native speaker thorough language revision and a more direct focus on the topic of the hypothesis on the mitochondrial defect origin and the use of natural drugs , after which revision it may be worth publishing.

Response: Thank you for your kind suggestion. We have conducted a thorough linguistic revision with AJE, and the following is our proof of revision (please see the attachment). The blue part of the text is the part of language revision carried out by a native speaker. The green part of the text is the part of the language revision carried out by the author. In addition, we have added the relevant content on the origin of mitochondrial defects in Lines 104-127 of the article and put forward the hypothesis of the use of natural drugs in Lines 458-472 of the article, hoping to meet your requirements.

lines 141-143– any reference?

Response: Thank you for pointing this out. We have checked and added references. The text you indicated is in Line 190 after the revision, and we have added reference [18] to this paragraph.

add space to line 120 "pnd delays the disease progression of obesity, T2DM, 121 NASH and other conditions[18]". there are numerous errors of this type in the text.

Response: We apologize for our mistakes. We have checked the full text, and added the missing spaces that precede the references in the full text. The text you indicated is in Line 163 after the revision. Thank you for your comments.

Comments on the Quality of English Language

language needs extensive editing.

Response: Thank you for your kind suggestion. We have revised the English grammatical errors and conducted a thorough linguistic revision with AJE.

see for example. line 30:

As diabetes mellitus is a major healthcare burden, the mechanism of type 2 diabetes (T2DM) has been widely discussed for many years, and the most important mechanism is insulin resistance.

Response: Thank you for your comments. The text you indicated is in Line 35 after the revision. We have changed “As diabetes mellitus is a major healthcare burden, the mechanism of type 2 diabetes (T2DM) has been widely discussed for many years, and the most important mechanism is insulin resistance” to “As type 2 diabetes (T2DM) is a major burden on healthcare systems, its mechanism has been widely studied for many years, and the most important mechanism leading to T2DM has been shown to be insulin resistance [2]”.

 a lot of clumsy sentences such as line 152: AGEs induce the expression of their own receptors  and bind with their receptors to activate NADPH

Response: Thank you for your comments. The text you indicated is in Line 198 after the revision. “AGEs induce the expression of their own receptors and bind with their receptors to activate NADPH, thereby promoting the formation of oxidative stress (OS), which in turn induces the expression of AGEs, thus aggravating the occurrence of mitochondrial stress” has been replaced by “The AGE-RAGE axis activates NADPH oxidases, which are the major endogenous sources of ROS, thus aggravating the occurrence of mitochondrial stress [37]”.

Line 180 For example, many scholars have proven that mi- roRNAs not only inhibit mitochondrial stress to improve insulin resistance but also directly reduce the effect of insulin resistance

Response: Thank you for your comments. The text you indicated is in Line 226 after the revision. We have changed “For example, many scholars have proven that microRNAs not only inhibit mitochondrial stress to improve insulin resistance but also directly reduce the effect of insulin resistance [29]” to “For example, many scholars have demonstrated that microRNAs can reduce insulin resistance either directly [43] or indirectly (by inhibiting mitochondrial stress and so on) [44]”.

Hard  to read text,  line 199: Mitochondrial stress disrupts the function of pancreatic beta cells, leading to impaired glucose tolerance and insulin resistance, but the relationship between mitochondrial stress and insulin resistance is certainly not through one factor, as many factors act  as links.

Response: Thank you for your comments. In fact, this passage is hard to read. Therefore, we added an additional paragraph describing the link between mitochondrial stress and insulin resistance and revised the sentences you pointed out in the hope that it would make the text easier to understand. The text is in Lines 245-260 after the revision.

Reviewer 2 Report

The article is of an overview nature and is devoted to substantiating the possible mechanism of action of components of plant origin that exhibit antidiabetogenic effects.The authors cite literature data, on the basis of which they express an opinion about the high role of oxidative stress of mitochondria in the development of glucose tolerance.The authors argue their position on the basis of modern literary data.At the same time, I would like to note that such a view is not new.The authors consider the problem of glucose tolerance very narrowly.It is known that many biologically active components of plants listed The authors have an antioxidant and antiradical effect, block inflammasomes and proinflammatory cytokines, prevent pathological glycation of long-lived proteins and neutralize glyoxal and methylglyoxal in diabetes mellitus.Therefore, the relationship between glucose tolerance and mitochondrial dysfunction may not be direct.

Author Response

Thank you for this valuable feedback. We have studied the comments carefully and made corrections in the manuscript according to your comments.

Response to Reviewer 2 Comments

The article is of an overview nature and is devoted to substantiating the possible mechanism of action of components of plant origin that exhibit antidiabetogenic effects.The authors cite literature data, on the basis of which they express an opinion about the high role of oxidative stress of mitochondria in the development of glucose tolerance.The authors argue their position on the basis of modern literary data. At the same time, I would like to note that such a view is not new.The authors consider the problem of glucose tolerance very narrowly.It is known that many biologically active components of plants listed The authors have an antioxidant and antiradical effect, block inflammasomes and proinflammatory cytokines, prevent pathological glycation of long-lived proteins and neutralize glyoxal and methylglyoxal in diabetes mellitus.Therefore, the relationship between glucose tolerance and mitochondrial dysfunction may not be direct.

Response: Thank you for your comments.

1. We have added the relevant content regarding the origin of mitochondrial defects in Lines 104-127 of the article and put forward the hypothesis of the use of natural drugs in Lines458-472 of the article. In addition, we have added some GDF15-related content in Lines 319-322 and Lines 325-329 of the article, which has not been summarized in previous reviews. We hope that this will improve the originality of the article.

2. Following your suggestion, we have revised the content related to glucose tolerance in Lines 45-49 of the article to make the description more objective.

3. We have added some extra content to describe the relationship between glucose tolerance and mitochondrial dysfunction in Lines 76-81 and Lines 245-260 of the article and pointed out that this complex relationship may stimulate more research ideas.

Round 2

Reviewer 1 Report

The Authors have answered to all of my concerns, therefore the mansucript is acceptable in its present form.